# The Advanced Care Study: Current Status of Lipedema in Spain, A Descriptive Cross-Sectional Study

**DOI:** 10.3390/ijerph20176647

**Published:** 2023-08-25

**Authors:** Alexo Carballeira Braña, Johana Poveda Castillo

**Affiliations:** Plastic Surgery Department, Lipedema Advanced Care, 46004 Valencia, Spain

**Keywords:** lipedema, lymphedema, painful adiposis, obesity, plastic surgery, liposuction, confinement

## Abstract

Introduction: The pathologic features of fatty tissue in lipedema are often challenging to diagnose, thus allowing for variable bias and leading to underdiagnosis. Lipedema is a disease that is currently little known worldwide, but it represents a public health problem and demands immediate, well-directed healthcare. Insufficient scientific information limits medical action, which limits making diagnoses and addressing an adequate multidisciplinary treatment. This study aims to evaluate the current state of lipedema in Spain to contextualize the disease’s pathophysiological characteristics and thus achieve a consensus that unifies and defines its diagnostic criteria and medical management. Likewise, this study aims to determine the effectiveness of the various treatments applied to the study patients and to evaluate the consequences of the pandemic related to this disease. Material and methods: The present work is a descriptive, cross-sectional study that analyzed online questionnaires. It was applied to 1069 patients and collected over 9 months between 2021 and 2022. The questionnaires were distributed to the leading national and regional associations of patients affected by lipedema. The study included all patients in a group who had a diagnosis of lipedema and in a group of undiagnosed patients with six or more symptoms. The variables analyzed were age, weight, height, body mass index (BMI), type of lipedema (according to Schingale’s classification), symptoms (according to Wolf’s classification, modified by Herbst), and treatments performed (physiotherapy, compression garments, sports, diet, radiofrequency, mesotherapy, and surgery), associated with the score given by the patients regarding the degree of improvement in their disease with each of these treatments. Results: There were 967 women and 2 men between 18 and 75 years old (mean of 38.5 years); a body weight between 33 and 150 kg (mean 75.8 kg); a height between 144 and 180 cm (mean 164 cm); and an average body mass index (BMI) of 28.1. The most common kind of lipedema in our study population was type III (affecting the hips, thighs, and calves). The treatment that individually improved patients’ quality of life the most was surgery, only surpassed by the multidisciplinary approach to the disease, including conservative measures. Conclusions: With this study, we can conclude that, in Spain, there is a real problem associated with the diagnosis of lipedema, specifying the need to seek this diagnosis actively and propose multidisciplinary management, since it offers the best overall results, of course not without forgetting that surgery is one of the most critical pillars in the approach to this disease. Consistent with the results obtained in this study, criteria were proposed and applied to represent a statistical value at the time of ruling on the clinical diagnosis of lipedema, considering that a patient who presents six or more of these diagnostic criteria, with a very high probability, will have lipedema.

## 1. Introduction and Objective

Lipedema is a chronic and progressive inflammatory disease of the loose connective tissue; it has an autosomal dominant inheritance of up to 60%; and its etiology is unclear. However, it is presumed to be associated with a compromise of the Aldo-Ketoreductase 1C1 (AKR1C1) gene [1]. It is characterized by a significant bilateral and symmetrical increase in the appendiceal volume of subcutaneous adipose tissue, giving it a nodular and fibrous appearance, predominantly in the buttocks, hips, and extremities [2,3]. In 30% of cases, it can also affect the upper extremities simultaneously. Its involvement is rare and definitely without the involvement of the hands, feet, or trunk [4,5,6].

It can become excruciating, a characteristic exacerbated by tactile pressure and a feeling of heaviness and exhaustion, which expresses a direct relationship associated with limitations in the personal and social development of patients [2,7]. In some cases, it reaches the development of lymphedema in advanced stages, establishing a marked limitation in mobility and, in turn, deteriorating the patient’s quality of life [2,4,8,9]. The resolution of these symptoms with the usual management of diet, exercise, and even bariatric surgery is unsuccessful, frequently causing frustration, eating disorders, and episodes of depression in this group of patients [6,10,11,12]. Likewise, it is important to mention the other characteristics of lipedema, such as the tendency to manifest ecchymosis, in some cases, the appearance of telangiectasias, cutaneous hypothermia, and the exacerbation of symptoms in the standing position [3,4,6,9,13].

Its diagnosis is purely clinical, supported by anamnesis, physical examination, and ruling out the differential diagnoses associated with lipedema [7,14,15]. In clinical practice, it can be challenging to differentiate lipedema from lymphedema because they coexist in the advanced stages of the disease. Therefore, it is necessary to consider signs such as the thickening of the skin, which is typical of lymphedema, and, on the other hand, the presence of pain upon pressure, non-pitting edema, a positive Cuff sign, and a negative Kaposi Stemmer sign, particularly in lipedema [15,16,17,18].

Its diagnostic confirmation can be supported by imaging techniques to assess the function of the lymphatic and vascular system in the lower extremities using ultrasonography, echo-Doppler, and lymphoscintigraphy more frequently, which have become valuable instruments in the differential diagnosis and equally valid as to stage the degree of involvement of the disease [19,20,21]. There are other diagnostic images such as Na-MRI (Sodium MRI), CT, NIRFLI (near-infrared fluorescence lymphatic imaging), SPY Elite, DEXA, Full Body Bioimpedance, Angiosterometer, and Streeten Test, which have also been used to clarify the diagnosis of lipedema; however, they lack notorious value judgment in routine clinical practice [19,22].

Considered by the World Health Organization (WHO) as a pathology belonging to painful adiposis syndrome and included since 2018 in the International Code of Disease (ICD 11—EF02.2), its incidence is commonly associated with microangiopathy, lymphatic disorders, and hormonal changes that include: puberty, pregnancy, and menopause, occurring almost exclusively in the female sex. Based on epidemiological data, it is estimated that it affects 12% to 18% of women and 0.2% of men worldwide, without a distinction of race [5,8,10].

The most used classifications at present are that of Schingale in 2003, typifying it according to the anatomical distribution of fat, and that of Meier Vollrath and Schmeller in 2004, who classified it according to the degree of commitment. In the development of this study, we used the first classification, since it includes more objective measurement variables at the time of physical examination, and in which four types of lipedema are described: type I, fatty tissue is concentrated around the hips and buttocks; type II, involvement from the hips to the knees; type III, alterations from the hips to the ankles, where the largest number of reported cases are grouped; and type IV, additional involvement of the upper extremities [11].

The scientific literature shows that there is no consensus focused on standardizing its symptoms and diagnostic criteria to confirm or rule out the presence of lipedema in suspected cases of patients with characteristics of the disease [8].

The treatment of lipedema is fundamentally aimed at treating pain and reducing the functional limitation caused by excessive volume in the affected areas, positively impacting patients’ quality of life. The systematic review describes a conservative management and a surgical one [22,23,24]. Among the conservative measures mentioned are decongestant therapy, the continuous use of compression stockings, diet, low-impact exercise, radiofrequency, and mesotherapy. For surgical management, the most frequently performed treatment is liposuction (WAL, PAL, UAL, lipolaser, and tumescent) in the calves, thighs, and arms [25,26,27,28].

## 2. Material and Methods

This is an observational, descriptive, cross-sectional study that responds to a non-experimental design. The data were obtained using an online questionnaire prepared by the authors as a tool, distributed through the different Spanish associations of patients with lipedema, both regional and national. In the questionnaire, 46 items were included: age, sex, weight, height, whether or not they belonged to any lipedema association, whether or not they were diagnosed (if the answer was positive, which specialist diagnosed them and how many medical visits were needed to obtain a confirmation of this diagnosis), and if not diagnosed, what the reason was for this.

Other information was obtained, such as family history, the stage of life in which they were diagnosed, whether or not they had healthcare, if they had diseases associated with lipedema, the degree of lipedema that was diagnosed, symptoms, and whether or not the pandemic worsened the progress of the lipedema and to what extent; they were also asked what type of treatment they had undergone (compression stockings, decongestant physiotherapy, anti-inflammatory diet, low-impact exercises, radiofrequency, mesotherapy, surgery, and others), to what degree each of the conservative measures and surgery improved their symptoms, and to what level the lipedema affected their personal role (quality of life). Regarding surgical management, details were asked, such as the areas of the body operated on (thighs, calves, armsm or various areas), what type of surgeon operated on them, the liposuction technique performed (WAL—Water Assisted Liposuction, PAL—Power Assisted Liposuction, tumescent, lipolaser, and vaser), how many times they underwent surgery, and in which areas.

The information was collected in the period between June 2021 and February 2022. The quantitative values were associated and interpreted in statistical tables for analysis and conclusion. A total sample of 1069 patients answered the questionnaire, to whom the inclusion criteria were applied. In the group of those not diagnosed, those with a number equal to or greater than 6 positive criteria of the 13 considered in the study and based on the proposed by Wolf and Herbst were included, thus finally delimiting an adequate population of (n = 969) patients distributed throughout the Spanish territory. Undiagnosed patients with less than six criteria present were excluded from the study.

The symptoms and criteria proposed in this study to diagnose patients with lipedema are illustrated in Table 1.

## 3. Results

The total number of patients who responded to the survey was 1069, 32.9% of which were not diagnosed with lipedema. Without standardized criteria and from the most conservative point of view, only those undiagnosed with six or more present symptoms could be considered as patients. Of this group of undiagnosed patients, 252 presented an average of 6.5 symptoms compared to 7.3 in the group of diagnosed patients; therefore, added to the 717 diagnosed, there was an adequate total sample of 969 study participants. See Table 2.

Based on the indicated weight, height, and BMI, it was estimated that 38.6% of the population was of an average weight, 31.9% were obese, 30.6% were overweight, and only 0.7% were underweight.

Of the adequate sample of 969 patients, 83.8% did not belong to an association for lipedema patients, while 16.2% did. Of this percentage, 26.8% belonged to ADALIPE, 23.6% to LIMFACALL, 21.7% to ACVEL, and the rest were distributed among different associations.

A vascular surgeon diagnosed 50.4% of the participants. In total, 51.2% required three or more visits to different specialists. In comparison, 33.7% did this more than five times to obtain a diagnosis, and of the group of undiagnosed participants, 50.8% expressed that they continued to actively seek a medical professional to diagnose them. A total of 52.1% had a confirmed family history, 28.8% believed that they did, and 18.8% did not. In total, 73.1% of the population responded that the disease developed at puberty, followed by 17% who stated that they had had lipedema since childhood, and 1.4% indicated that it appeared in menopause. A total of 40.9% had public healthcare, 21.9% private, 37% had both, and only 2% did not have healthcare coverage. Regarding the development of lipedema in association with other diseases, 51.6% of the sample answered yes, relating its manifestation in 28.1% with a history of obesity, 15.8% with hypothyroidism, 8.6% with lymphedema, 0.8% with type II diabetes mellitus, and the remaining percentage was associated with other diseases. In total, 41.7% of patients claimed to have type III lipedema, followed by 36.8% with type IV, 17.8% with type II, and 3.7% with type I. See Figure 1.

According to the presentation of symptoms, the feeling of heaviness or swollen legs obtained the highest score, with 94.1% of the surveyed population, followed by 91.7% for non-response to diets, 88.3% for the tendency to bruise, 86% for non-response to exercise, a volume disproportion between the arms and legs, as well as pain on palpation with 83%, 81.4% denied the involvement of the hands and feet, 75.9% stated that they had a stiff consistency and nodules in the affected areas, 55.5% answered with spontaneous pain, and 6% denied presenting symptoms. The weight gain during confinement due to the pandemic was 3 kg, and 64% of those surveyed stated that their symptoms worsened with an average of 6 points out of 10, while the remaining 36% stated that they were unaffected. In total, 53.8% attended a follow-up medical consultation for the disease last year.

A total of 59.1% of the study participants followed a diet distributed as follows: 25% a ketogenic diet (keto, protein, and carbohydrate-free), 17.2% an anti-inflammatory diet (gluten-free and dairy-free), and 15% followed a healthy diet (fruits, vegetables, and Mediterranean), while 9.4% of the respondents followed others such as vegetarian, fasting, and hypocaloric.

In relation to the difficulties that lipedema represents in daily life, the results are illustrated in Table 3.

Regarding the improvement in symptoms, the questions were evaluated on a scale from 0 to 10 points, and these were the results: with diet, the average was 6.1, with exercise, the average was 5.5, with an average for mean compression therapy of 5.8, for decongestive physiotherapy an average of 6.3, an average for radiofrequency of 3.7, an average for mesotherapy of 3.5, and with surgery, an average of 7.8 points, with conjugate therapy being the one with the highest score. The percentage relationship is represented in Figure 2.

Eighty five percent of the diagnosed patients underwent surgery. The calves were the most intervened body area with 62.8%, followed by the thighs at 57.8%, arms at 16.7%, various areas at 11.7%, and unspecified areas at 8.9%. In total, 44.4% of the surveyed population stated that they did not know the specialty of the doctor who operated on them, 35.6% asserted that a plastic surgeon operated on them, 18.9% were operated on by a general surgeon, 6% by a dermatologist, and another 6% by an aesthetic doctor. The liposuction technique in the sample was WAL at 65.4%, followed by 17% with tumescent liposuction, 11.1% with vaser, another 11.1% with PAL, and lipolaser at 9.8%. Regarding the degree of improvement according to the type of liposuction performed, on a scale of 0 to 10, with an average of 8.6, PAL obtained the highest score, followed by 8.5 for WAL, 7.8 for tumescent, an average of 5.5 for vaser, and 4.5 for lipolaser. See Table 4.

Regarding the number of times that they underwent surgery, 53.3% did so once, 28.9% underwent surgery twice, or more times in different body areas, 12.2% did not specify their answer, and 5.6% did twice, or more times in the same area.

## 4. Discussion

Lipedema is a disease of recent clinical recognition. Several authors have described in their works concepts that have made it possible to expand the existing scientific information, motivating interest in continued research focused on centralizing and standardizing the specific aspects aimed at the diagnosis and medical–surgical management of this disease [8].

This review had an inclusion rate of 89.65%, in which all the participants were assessed without differentiation, eliminating selection bias and confounding factors. The analysis used Pearson correlation, demonstrating a correlation between the number of symptoms present and the diagnosis. By applying the non-parametric Mann–Whitney test, we demonstrated the distribution of the number of symptoms in the diagnosed group. In the undiagnosed group, it was similar (*p* = 0.666), representing a high statistical reliability of the results, allowing us to conclude that, on average, six or more symptoms described and proposed in this study were enough to diagnose a patient with lipedema.

A significant percentage of the sample showed that, to obtain a diagnosis, an active search was necessary that included several visits to different medical specialists; this is the reason why we included in the study undiagnosed patients with six or more present symptoms, considering them with a high probability as patients with lipedema.

The autosomal dominant inheritance of 60% for lipedema described in the literature is correlated with the conclusive results obtained in this study concerning the genetic background of this population group.

Other results, such as the high prevalence of the disease in women, with them being 99.8% of the sample, coincides with the information found in the reviewed literature.

No clear association of hypothyroidism with lipedema was found; however, the prevalence of diabetes mellitus in lipedema is striking, showing a shallow relationship concerning the general Spanish population of women of any age, 6.06% versus 0.8% of the population of this study. However, in this population group, the prevalence of being overweight or obese was higher compared to the Spanish adult population, which was 53.60% (61.40% of men and 46.10% of women) [29,30], making it possible to highlight that the peripheral accumulation of fat that is characteristic of lipedema can act as a positive effect regarding the predisposition of insulin resistance described in the article “Lipedema: friend and foe” [14].

It is essential to highlight that the incidence of lymphedema in this population was four times higher than that in the general Spanish population. These data are proportional to what is reported in the literature reviewed in this study.

The diets that produced the best results were ketogenic and anti-inflammatory, a critical review when determining an individualized nutritional guideline and, added to the other treatment items, provide a patient with the tools to stop the progression of their symptoms.

The conservative treatment group had an average of 5.5 points out of 10, with mesotherapy and radiofrequency being the ones that seemed to work the least. Surgery was the treatment that yielded the best results. However, a combination of all the treatments exceeded the level of improvement, with 95.3% (40.6% radical improvement and 54.7% partial improvement).

As we have mentioned in previous sections, the absence of specific clinical parameters regarding lipedema limits medical action in the multidisciplinary field, neglecting the needs present in this group of patients that is increasing in numbers, making it a severe public health problem at the global level. Factors such as a lack of scientific bases that allow for controlling the progression of lipedema could be reflected in the advance of the functional limitations and deterioration of the quality of life of a population group that, based on the age of the development of the disease, are in a potentially productive period [7].

During the statistical analysis of our database, we found that the variables of sex, age, and family history were decisive in the context of disease development, which means that there was a linear correlation between these variables and lipedema, with a great magnitude of prediction.

From the multivariate analysis, we note that a vascular surgeon made the diagnosis in most cases and this was achieved, on average, with three or more consultations with different specialists. We verified that most of the treatments implied a rate of radical reduction in symptoms of 25–30%, with surgery having the highest score, except the joint application of all the treatments, which raised it to 40.6%. The most frequent liposuction technique for the patients with lipedema was WAL, performed in a higher percentage by a plastic surgeon. The majority of the surveyed population had been operated on only once. In this sense, the rate of surgical reintervention in the same area was meagre, at 5.6%, demonstrating the effectiveness and stability of surgical management for this group of patients.

## 5. Conclusions

We verified that the confinement secondary to the pandemic in this group of patients exacerbated their symptoms and facilitated the progression of the disease.

At present, and despite the high prevalence of lipedema, achieving a diagnosis represents a real problem that limits the early initiation of its treatment; this is due to the need for a greater consensus regarding the number of diagnostic criteria necessary to confirm it.

The statistical analysis of the data obtained in this study allowed us to conclude that individuals with six or more symptoms, based on those proposed by Wolf and Herbst, with a high probability, were patients with lipedema.

Even though in this disease, the prevalence of being overweight or obese is higher than that in the general population, its association with diabetes mellitus is much lower, since insulin resistance is associated with fat distribution at the central level. At the same time, lipedema is characterized by its presence at the peripheral level.

Surgical treatment as an isolated measure was the one that offered the best results, only being surpassed by its combined application with the other therapeutic measures studied.

## Figures and Tables

**Figure 1 ijerph-20-06647-f001:**
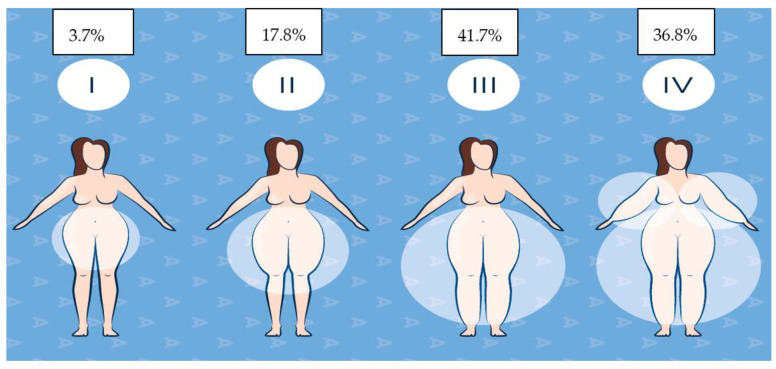
Percentage of patients according to the Schingale classification.

**Figure 2 ijerph-20-06647-f002:**
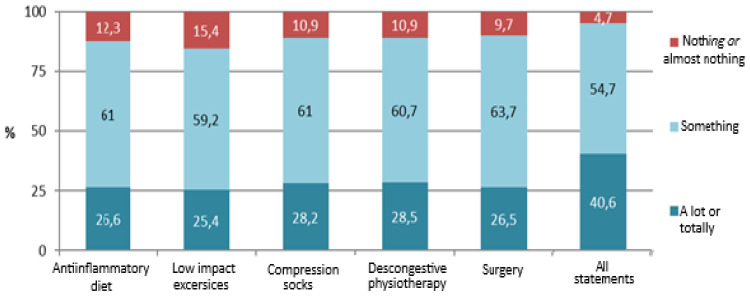
Improvement of symptoms according to treatment.

**Table 1 ijerph-20-06647-t001:** Symptoms and criteria for diagnosing lipedema.

	Is Diagnosed					
	Total		No		Yes	
	N	%	N	%	N	%
Total	1069	100%	352	100%	717	100%
Feeling of heaviness or swollen legs	989	92.50%	318	93.30%	671	93.60%
No response to diet: minimal loss in the legs and arms	951	89%	305	86.60%	646	90.10%
Tendency to bruise: frequently in legs without origin or minor trauma	909	85%	283	80.40%	626	87.30%
No response to physical exercise: minimal loss of volume in legs and arms	887	83%	276	78.40%	611	85.20%
Pain on palpation: painful sensation before slight stimuli in the legs	839	76.50%	250	71%	589	82.10%
Presents a clear disproportion of volume in the legs versus the arms and trunk	836	78.20%	245	69.60%	591	82.40%
Unaffected hands and feet	817	76.40%	243	69%	574	80.10%
Hard and nodular consistency in the fat of the legs and arms	757	70.80%	222	63.10%	535	74.60%
Spontaneous pain in arms and legs despite rest	554	51.80%	154	43.80%	400	55.80%
None	7	0.70%	1	0.30%	6	0.80%

**Table 2 ijerph-20-06647-t002:** Percentage of diagnosed patients.

	N	%
Total	1069	100%
Not	352	39.90%
Yes	717	67%

**Table 3 ijerph-20-06647-t003:** What difficulties does lipedema represent?

	N	%
Total	926	100%
Difficulty finding clothes	758	91.90%
Restriction to perform certain activities and sports	676	73%
Difficulty in socializing	386	41.70%
Difficulty performing daily tasks	336	36.30%
Difficulty at the level of a romantic partner	322	34.80%
Others	213	23%
Difficulty at the family level	153	16.50%
Difficulty finding suitable Jobs	66	7.10%

**Table 4 ijerph-20-06647-t004:** Improvement of symptoms according to liposuction technique.

	Type of Liposuction					
	Total	WAL (Water-Assistant Lip)	Tumescent	PAL (Power Assistant Lip)	Vaser	Lipolaser
N	147	98	24	18	13	13
Medium	7.8	8.5	7.8	8.6	5.5	4.5
Typical deviation	2.4	1.6	2.5	1.7	3.5	2.6
Minimum	0	0	0	5	0	0
Maximum	10	10	10	10	10	10
Median	8	9	8	9	6	5

## Data Availability

The data presented in this study are available upon request from the corresponding author. The data is not publicly available due to future projects based on the information obtained in this study.

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
