# Peer review of "The Advanced Care Study: Current Status of Lipedema in Spain, A Descriptive Cross-Sectional Study"

_ijerph, 2023, doi:10.3390/ijerph20176647_

Round 1
Reviewer 1 Report
Abstract:
Please shorten the abstract to the most relevant facts for each section.
Introduction:
Lipedema is a chronic and progressive inflammatory disease of loose connective tis-sue; it has an autosomal dominant inheritance of up to 60%;
--> Please add a reference for this.
The resolution of these symptoms with the usual management of diet, exercise and even bari-atric surgery is unsuccessful, frequently causing frustration, eating disorders and epi-sodes of depression in this group of patients
--> Bariatric surgery can also help in Lipedema patients. Please add this reference: Fink, J. M., Schreiner, L., Marjanovic, G., Erbacher, G., Seifert, G. J., Foeldi, M., & Bertsch, T. (2021). Leg volume in patients with lipoedema following bariatric surgery. Visceral Medicine, 37(3), 206-211.
Results:
Based on the indicated weight, height and BMI, it was estimated that 38.6% of the population is of average weight, 31.9% are obese, 30.6% are overweight, and only 0.7% are underweight.
--> Please state the BMI for each group.
Of the adequate sample of 969 patients, 83.8% did not belong to an association for lipedema patients, while 16.2% did. Of this percentage, 26.8% belonged to ADALIPE, 23.6% to LIMFACALL, 21.7% to ACVEL and the rest distributed among different associ-ations.
--> Please write the full names of the associations.
A vascular surgeon diagnosed 50.4% of the participants.
--> Please rank the other specialists as well: plastic surgeons, dermatologists ....
A vascular surgeon diagnosed 50.4% of the participants. 51.2% required 3 or more visits to different specialists. In comparison, 33.7% did it more than 5 times to obtain the diagnosis and of the group of undiagnosed 50.8% expressed that they continue to seek a medical professional actively.
--> This sentence sounds weird, please phrase it more clear.
33.7% did it more than 5 times to obtain the diagnosis and of the group of undiagnosed 50.8% expressed that they continue to seek a medical professional actively. To diagnose them.
--> Please check the punctuation.
73.1% of the population responded that the disease developed at puberty, followed by 17%
--> if you decide to use one number behind the dot, so use it everywhere: 17.0% for example
40.9% have public healthcare, 21.9% private, 37% have both, and only 2% do not have healthcare coverage.
--> same here
Regarding the improvement of symptoms, the questions were evaluated on a scale from 0 to 10 points, and these are the results: with the diet, the average was 6.1; with exercise, the average was 5.5, with the averages mean compression therapy of 5.8, for de-congestive physiotherapy an average of 6.3, average radiofrequency of 3.7, average mes-otherapy of 3.5 and with surgery an average of 7.8 points, conjugate therapy being the one with the highest score. The percentage relationship is represented in Table 4.
--> I don't understand the scale and the meaning of the numbers. Please explain.
--> Check spellings in Table 4.
Eighteen five percent of the diagnosed patients underwent surgery.
--> Please write 18.5 %
Regarding the degree of improvement according to the type of liposuction performed, on a scale of 0 to 10 with an average of 8.6, Int. J. Environ. Res. Public Health 2021, 18, x FOR PEER REVIEW 6 of 8
PAL obtained the highest score, followed by 8.5 for WAL, 7.8 for tumescent, an average of 5 .5 for vaser and 4.5 for lipolaser. See table 5.
-->I don't understand. Please explain. Also table 5.
--> The questionnaire should be provided.
Author Response
Open Review ( ) I would not like to sign my review report
(x) I would like to sign my review report Quality of English Language ( ) English very difficult to understand/incomprehensible
(x) Extensive editing of English language and style required
( ) Moderate English changes required
( ) English language and style are fine/minor spell check required
( ) I am not qualified to assess the quality of English in this paper
| Yes | Can be improved | Must be improved | Not applicable | |
| Does the introduction provide sufficient background and include all relevant references? | ( ) | ( ) | (x) | ( ) |
| Are all the cited references relevant to the research? | ( ) | (x) | ( ) | ( ) |
| Is the research design appropriate? | ( ) | (x) | ( ) | ( ) |
| Are the methods adequately described? | ( ) | (x) | ( ) | ( ) |
| Are the results clearly presented? | ( ) | ( ) | (x) | ( ) |
| Are the conclusions supported by the results? | ( ) | ( ) | (x) | ( ) |
Comments and Suggestions for Authors
Abstract:
Please shorten the abstract to the most relevant facts for each section.
Introduction:
Lipedema is a chronic and progressive inflammatory disease of loose connective tis-sue; it has an autosomal dominant inheritance of up to 60%;
--> Please add a reference for this.
The resolution of these symptoms with the usual management of diet, exercise and even bari-atric surgery is unsuccessful, frequently causing frustration, eating disorders and epi-sodes of depression in this group of patients
--> Bariatric surgery can also help in Lipedema patients. Please add this reference: Fink, J. M., Schreiner, L., Marjanovic, G., Erbacher, G., Seifert, G. J., Foeldi, M., & Bertsch, T. (2021). Leg volume in patients with lipoedema following bariatric surgery. Visceral Medicine, 37(3), 206-211.
Results:
Based on the indicated weight, height and BMI, it was estimated that 38.6% of the population is of average weight, 31.9% are obese, 30.6% are overweight, and only 0.7% are underweight.
--> Please state the BMI for each group.
Of the adequate sample of 969 patients, 83.8% did not belong to an association for lipedema patients, while 16.2% did. Of this percentage, 26.8% belonged to ADALIPE, 23.6% to LIMFACALL, 21.7% to ACVEL and the rest distributed among different associ-ations.
--> Please write the full names of the associations.
A vascular surgeon diagnosed 50.4% of the participants.
--> Please rank the other specialists as well: plastic surgeons, dermatologists ....
A vascular surgeon diagnosed 50.4% of the participants. 51.2% required 3 or more visits to different specialists. In comparison, 33.7% did it more than 5 times to obtain the diagnosis and of the group of undiagnosed 50.8% expressed that they continue to seek a medical professional actively.
--> This sentence sounds weird, please phrase it more clear.
33.7% did it more than 5 times to obtain the diagnosis and of the group of undiagnosed 50.8% expressed that they continue to seek a medical professional actively. To diagnose them.
--> Please check the punctuation.
73.1% of the population responded that the disease developed at puberty, followed by 17%
--> if you decide to use one number behind the dot, so use it everywhere: 17.0% for example
40.9% have public healthcare, 21.9% private, 37% have both, and only 2% do not have healthcare coverage.
--> same here
Regarding the improvement of symptoms, the questions were evaluated on a scale from 0 to 10 points, and these are the results: with the diet, the average was 6.1; with exercise, the average was 5.5, with the averages mean compression therapy of 5.8, for de-congestive physiotherapy an average of 6.3, average radiofrequency of 3.7, average mes-otherapy of 3.5 and with surgery an average of 7.8 points, conjugate therapy being the one with the highest score. The percentage relationship is represented in Table 4.
--> I don't understand the scale and the meaning of the numbers. Please explain.
--> Check spellings in Table 4.
Eighteen five percent of the diagnosed patients underwent surgery.
--> Please write 18.5 %
Regarding the degree of improvement according to the type of liposuction performed, on a scale of 0 to 10 with an average of 8.6, Int. J. Environ. Res. Public Health 2021, 18, x FOR PEER REVIEW 6 of 8
PAL obtained the highest score, followed by 8.5 for WAL, 7.8 for tumescent, an average of 5 .5 for vaser and 4.5 for lipolaser. See table 5.
-->I don't understand. Please explain. Also table 5.
--> The questionnaire should be provided.
Submission Date 12 March 2023 Date of this review 22 Mar 2023 09:45:15 © 1996-2023 MDPI (Basel, Switzerland) unless otherwise stated
Reviewer 2 Report
Introduction :
The article by Michelini S, et al.. ‘Aldo-Keto Reductase 1C1 (AKR1C1) as the First Mutated Gene in a Family with Nonsyndromic Primary Li-pedema. Int J Mol Sci. 2020) reported only one familiy. The authors should state that other factors including genetics, may be associated/causative of familial occurrence of lipedema.
I agree that diagnosis of lipedema is difficult as it is often dismissed with obesity and lymphedema. Ultrasound is useful as stated by the authors to assess the diagnostic. I suggest to cite the work by Naouri M et al. High-resolution cutaneous ultrasonography to differentiate lipoedema from lymphoedema. Br J Dermatol 2010; 163: 296–301. Indeed, dermal edema is usually absent at least at early stage, contrary to lymphedema.
When edema is present, congestive therapy and decongestant therapy may be useful. The authors should cite « Atan T, Bahar-Özdemir Y. The Effects of Complete Decongestive Therapy or Intermittent Pneumatic Compression Therapy or Exercise Only in the Treatment of Severe Lipedema: A Randomized Controlled Trial. Lymphat Res Biol. 2021 Feb;19(1):86-95. » as this is a randomized controlled study.
Ref 3 : the title is lacking, and available in German. I suggest to replace it with a widely available and actualized reference « Reich-Schupke S, Schmeller W, Brauer WJ, Cornely ME, Faerber G, Ludwig M, Lulay G, Miller A, Rapprich S, Richter DF, Schacht V, Schrader K, Stücker M, Ure C. S1 guidelines: Lipedema. J Dtsch Dermatol Ges. 2017 Jul;15(7):758-767. doi:10.1111/ddg.13036. PMID: 28677175.
At the end of the discussion, I suggest to add the objectives of the study.
Table 1 :
« Uninfected hands and fee » to be replaced with unaffected (or spared ?)
Discussion : can the authors summarize at the beginning of the discussion the main faindings from their work ?
Conclusion :
The author state : « We verified that the confinement secondary to the pandemic in this group of patients exacerbated the symptoms and facilitated the progression of the disease ». How did they do ? I don’t find anything in the results.
Author Response
Open Review
(x) I would not like to sign my review report
( ) I would like to sign my review report
Quality of English Language
( ) English very difficult to understand/incomprehensible
( ) Extensive editing of English language and style required
( ) Moderate English changes required
(x) English language and style are fine/minor spell check required
( ) I am not qualified to assess the quality of English in this paper
| Yes | Can be improved | Must be improved | Not applicable | |
| Does the introduction provide sufficient background and include all relevant references? | ( ) | (x) | ( ) | ( ) |
| Are all the cited references relevant to the research? | ( ) | (x) | ( ) | ( ) |
| Is the research design appropriate? | (x) | ( ) | ( ) | ( ) |
| Are the methods adequately described? | ( ) | (x) | ( ) | ( ) |
| Are the results clearly presented? | ( ) | (x) | ( ) | ( ) |
| Are the conclusions supported by the results? | ( ) | (x) | ( ) | ( ) |
Comments and Suggestions for Authors
Introduction :
The article by Michelini S, et al.. ‘Aldo-Keto Reductase 1C1 (AKR1C1) as the First Mutated Gene in a Family with Nonsyndromic Primary Li-pedema. Int J Mol Sci. 2020) reported only one familiy. The authors should state that other factors including genetics, may be associated/causative of familial occurrence of lipedema.
I agree that diagnosis of lipedema is difficult as it is often dismissed with obesity and lymphedema. Ultrasound is useful as stated by the authors to assess the diagnostic. I suggest to cite the work by Naouri M et al. High-resolution cutaneous ultrasonography to differentiate lipoedema from lymphoedema. Br J Dermatol 2010; 163: 296–301. Indeed, dermal edema is usually absent at least at early stage, contrary to lymphedema.
When edema is present, congestive therapy and decongestant therapy may be useful. The authors should cite « Atan T, Bahar-Özdemir Y. The Effects of Complete Decongestive Therapy or Intermittent Pneumatic Compression Therapy or Exercise Only in the Treatment of Severe Lipedema: A Randomized Controlled Trial. Lymphat Res Biol. 2021 Feb;19(1):86-95. » as this is a randomized controlled study.
Ref 3 : the title is lacking, and available in German. I suggest to replace it with a widely available and actualized reference « Reich-Schupke S, Schmeller W, Brauer WJ, Cornely ME, Faerber G, Ludwig M, Lulay G, Miller A, Rapprich S, Richter DF, Schacht V, Schrader K, Stücker M, Ure C. S1 guidelines: Lipedema. J Dtsch Dermatol Ges. 2017 Jul;15(7):758-767. doi:10.1111/ddg.13036. PMID: 28677175.
At the end of the discussion, I suggest to add the objectives of the study.
Table 1 :
« Uninfected hands and fee » to be replaced with unaffected (or spared ?)
Discussion : can the authors summarize at the beginning of the discussion the main faindings from their work ?
Conclusion :
The author state : « We verified that the confinement secondary to the pandemic in this group of patients exacerbated the symptoms and facilitated the progression of the disease ». How did they do ? I don’t find anything in the results.
Submission Date
12 March 2023
Date of this review
28 Mar 2023 11:35:06
© 1996-2023 MDPI (Basel, Switzerland) unless otherwise stated